# Body Image Assessment Tools in Pregnant Women: A Systematic Review

**DOI:** 10.3390/ijerph20032258

**Published:** 2023-01-27

**Authors:** Eduardo Borba Salzer, Juliana Fernandes Filgueiras Meireles, Alesandra Freitas Ângelo Toledo, Marcela Rodrigues de Siqueira, Maria Elisa Caputo Ferreira, Clara Mockdece Neves

**Affiliations:** 1Faculdade de Educação Física e Desportos, Universidade Federal de Juiz de Fora, Juiz de Fora 36036-900, Brazil; 2Department of Family and Community Medicine, School of Community Medicine, University of Oklahoma, Tulsa, OK 74120, USA

**Keywords:** body image, pregnancy, validation studies, psychometrics

## Abstract

Pregnancy is a remarkable time and generates several changes in women in a short period. Body image is understood as the mental representation of the body itself, and, although bodily changes are considered healthy, they can impact pregnant women’s body image. Problems related to body image during pregnancy can affect the health of the mother and fetus; thus, it is essential for health professionals to detect potential disorders as soon as possible. The objective of this systematic review was to identify instruments for assessing body image in pregnant women, highlighting their main characteristics. To this end, we applied the recommendations of the Preferred Reporting Items for Systematic Reviews and Meta-Analyses to searches in the EMBASE, PubMed, and American Psychological Association databases from 5 January to 10 August 2021. We included studies on adult pregnant women without comorbidities in the validation and adaptation of (sub)scales that analyze components of body image. We excluded studies that considered nonpregnant, adolescent, postpartum, and/or clinical populations, as well as smoking/drug use studies that were not validation studies or did not assess any aspect of body image. We investigated the quality of the studies using the Quality Assessment Tool for Studies with Diverse Designs. In all, we examined 13 studies. The results point to a growing concern over body image during pregnancy, as there has been an increase in the number of validation and adaptation studies involving scales for different cultures that scrutinize different constructs. The findings suggest that the listed instruments be used in future research.

## 1. Introduction

Pregnancy is a period in which a fetus developed inside the uterus of a woman [1]. It is a remarkable and highly complex period in women’s lives, bringing about several biopsychosocial changes in a short time of approximately 40 weeks [2,3,4]. Although expected, these changes can impact pregnant women’s body image [5,6].

Body image is a mental representation of the body itself, which can be influenced by physiological, sociological, and libidinal aspects [7]. Body image is changeable and complex [8,9], and, throughout pregnancy, women tend to re-evaluate it [9,10].

A negative body image during pregnancy can negatively impact women’s health and wellbeing [10,11]. Body dissatisfaction during pregnancy is related to eating disorders [12], emotional instability, anxiety, and depression [1,13], with negative consequences for the fetus, such as low birth weight and low breastfeeding rates [6]. Meireles et al. [14] discovered controversies and unclear outcomes regarding body dissatisfaction in pregnant women. Some authors have found increasing satisfaction throughout pregnancy [2,13,15]. However, studies have shown high levels of pregnant women dissatisfied with their appearance [16,17], achieving levels of 45% [18]. These discrepancies may be explained due to the different methods of body image’s evaluation [3,14].

In addition, there is a scarcity of body image assessment tools for pregnant women [3,14,19]. In realizing the importance of choosing appropriate tests to guarantee the quality of an investigation [20,21], it is crucial to analyze the current scenario of instrument validation for pregnant women’s body image.

We aimed to identify existing body image assessment instruments for pregnant women, pointing out their main characteristics, through a systematic review of the literature.

## 2. Materials and Methods

We performed a systematic review following the recommendations of the Preferred Reporting Items for Systematic Reviews and Meta-Analyses (PRISMA) [22]. We sought to compile existing instruments at the global level for the assessment of body image in pregnant women, examining their psychometric features, the constructs on which they focus, and the populations on which they were normed. As such, we aimed to provide greater clarity on which instruments and tests have been developed and validated for pregnant women. We registered this research with Prospero under the code CRD42021264801.

### 2.1. Search Strategy

We carried out searches in the EMBASE, PubMed, and American Psychological Association (APA) databases from 5 January to 10 August 2021. For this, we used the patient, intervention, comparison, and outcome (PICO) strategy to formulate the research question, and we chose the search words via the mesh descriptors and their respective registration terms. We used a sample of pregnant women as the population; the intervention refers to the variable “body image” and the outcome refers to “validation” or “psychometric” studies using the Boolean operator AND between each word group. In addition, we performed manual searches of the references of cited articles in the gray literature. Table 1 outlines the complete search strategy.

### 2.2. Selection of Studies

Two researchers independently carried out the search, selection, and analysis of the articles, while a third researcher was responsible for examining any doubts. As criteria for the inclusion of articles, we used (1) articles whose objective was to describe the development or cross-cultural adaptation of scales for assessing body image, and (2) studies carried out with a population of pregnant women without any comorbidities. We decided to not limit the range of publication dates, including articles published in any period.

The filters used in the search were “full text available” and articles written in English and Portuguese. Because several investigations on ultrasounds and animal pregnancy appeared, we chose to mark the filter “human”. Figure 1 presents the entire selection and refinement process of the studies included in the systematic review.

After the search, we exported all identified articles to EndNote Web software and excluded duplicates. Initially, we filtered eligible articles by title, followed by abstract, and finally by full text analysis.

### 2.3. Exclusion Criteria

To select articles, we excluded studies that (1) considered nonpregnant women, (2) contained samples of pregnant adolescents, (3) evaluated postpartum women, (4) included clinical populations, (5) were related to smoking and/or drug use, (6) did not present scale validation data, and (7) did not assess any aspect of body image.

### 2.4. Data Extraction

Initially, we created a database spreadsheet. One researcher performed data extraction and analysis, which a second researcher subsequently verified.

We extracted the following data from the texts: (1) complete reference (year of publication); (2) sample characteristics; (3) measurement instruments used in the study; (4) main results; (5) psychometric tests used; (6) quality of the included studies.

### 2.5. Quality Analysis of the Studies

Two researchers independently analyzed the quality of the articles, and a third researcher helped to resolve any differences. To this end, we used the Quality Assessment Tool for Studies with Diverse Designs (QATSDD) [23]. The QATSDD evaluates both quantitative studies (14 items) and qualitative studies (14 items) and may also consider mixed-methods studies (16 items). For this research, we used the 14 assessment items for quantitative methods, scoring them from 0 to 3, assigning 0 points when the author did not mention anything regarding the analyzed category, 1 point when a little information was stated, 2 points when the information was provided in some way, and 3 points when the information was presented accurately.

We scrutinized the studies’ quality based on the calculation of the maximum percentage achieved (42 points). We considered a cutoff point of 50%, with values above that indicating good quality and scores below that denoting lower quality than expected [23].

## 3. Results

We identified 946 studies in the databases and excluded 236 duplications and 541 articles after applying the filters. After reading the studies in full, we further excluded 161 studies because they did not meet the inclusion criteria, leaving eight articles for which we analyzed their results through the databases.

In addition to the studies found in the initial search, we manually added five articles as they did not appear directly while searching the databases. After the entire process, we scrutinized 13 studies in total (see Figure 1).

Table 2 presents the characteristics of the 13 selected studies. Regarding location, our results highlight a global concern about the assessment of pregnant women’s body image when identifying studies conducted on different continents: North America [24,25], South America [26,27], Europe [3,11,28,29,30], Asia [31,32,33], and Oceania [34]. The selected articles were from Australia, the United Kingdom (UK), Brazil, Germany, Israel, Japan, Turkey, the United States of America (USA), and Iran, along with a multicenter study that examined pregnant women in the USA and Canada. The exceptions were Africa and Central America, which did not present any validation studies or scale creation for pregnant women.

Of the selected articles, seven entailed (sub)scales that evaluate some construct of pregnant women’s body image, four studies involved scale validations for another country, an instrument validation study investigated a sample of adult pregnant women, and another instrument assessed body dissatisfaction through a silhouette scale.

The studies included a sample size range from 161 [33] to 1288 [26] pregnant women aged between 18 to 52 years old.

Regarding the number of items of the identified instruments, the lowest number of items was identified in the Self-Acceptance Scales for Pregnant Women (SAS-PW) [26] and Pregnancy-related Anxiety Questionnaire-Revised 2 (PRAQ-R2) [29] with 10 items. While the first instrument [26] is composed of two subscales, the second tool [29] evaluates anxiety and has only one subscale related to body image. In addition, one study used a silhouette scale with two items to verify dissatisfaction with body size during pregnancy [33]. The instrument with the highest number of items was the Childbearing Attitudes Questionnaire (CAQ) [25], which presented 73 items subdivided into 16 factors.

This systematic review identified a diversity of aspects of body image considered by the instruments, including body dissatisfaction, attractiveness, concerns about fat or weight gain, concerns about physical appearance, pregnant appearance, body and facial features, and body satisfaction.

There was a predominance of cross-sectional studies (12 studies), with a concern in the evaluation of the three gestational trimesters (nine studies), and the main measure of internal consistency was Cronbach’s α (12 studies). Out of the 13 studies, seven of them performed both EFA (exploratory factor analysis) and CFA (confirmatory factor analysis), four ran EFA, and one analyzed the items using only CFA. In addition, eight studies performed convergent validity.

The results of QATSDD tool showed adequate outcomes in terms of quality for all the 13 articles included in this review (Table 2).

## 4. Discussion

Body image must be analyzed during pregnancy in order to promote mental health [21]. Kirk and Preston [3] and Meireles et al. [14] pointed out that the findings on pregnant women’s body image are still very controversial because many studies involve instruments that have not been validated for the target audience in question. Hence, we aimed to establish which instruments have been validated to evaluate pregnant women’s body image, as well as to pinpoint the aspects assessed by them. As a result, we identified 13 studies that met the inclusion criteria and which we subsequently carefully analyzed. We found seven questionnaires that assessed pregnant women’s body image [3,24,25,26,28,31,32], an instrument developed for the adult population and adapted for pregnant women [34], four articles on adapting scales to other cultures [11,27,29,30], and an instrument that assesses body dissatisfaction through a silhouette scale [33].

Some authors indicated that most measures used to assess body image in pregnant women are adaptations of measures for other audiences [3,14,19]. Thus, our results point to a growing concern among researchers regarding the creation of new instruments that seek to specifically assess body image in women during this very complex time in their lives [3,24,26,28,29,31,32,33,34]. This is a relevant finding, as the creation of scales that consider the reality and specificity of the target population substantially increases the chances that the information collected will express what is desired to be measured.

In addition to the existing concern surrounding the creation of new measures, the findings underscore researchers’ interest in adapting instruments to other cultures. In all, we identified four studies that adapted scales for other countries [11,27,29,30]. Of these, two studies adapted the Body Image in Pregnancy Scale (BIPS) for Germany [30] and Brazil [27]. The Body Understanding Measure for Pregnancy Scale (BUMPs) was adapted for Turkey pregnant women [11]. Furthermore, one study carried out the Body Attitudes Questionnaire (BAQ) validation process for Australian pregnant women [34].

The findings indicate a variety of constructs used, with a view to assessing pregnant women’s body image, with an emphasis on body dissatisfaction [24,27,30,32,33,34], dissatisfaction with one’s attractiveness [24,27,30,32], concerns regarding fat or weight gain [3,11,28,32,34], concerns about one’s physical appearance, pregnant appearance, or body and facial features [3,11,24,25,26,27,28,29,30,31], acceptance of pregnancy [26], and body satisfaction [3,11]. Since body image is a multidimensional concept, it is important to include a broad range of dimensions in research that are relevant to the analyzed construct. Hence, our findings present a diversity of dimensions assessed, making it necessary to understand them and to use them together to minimize errors in the examination of the results.

Ferreira et al. [35] (p. 28) underlined the importance of “establishing a relationship of temporal precedence between the characteristics that are associated with the dimensions of body image”. However, regarding the design of the studies, we noted a predominance of cross-sectional investigations to the detriment of longitudinal ones. Only Fuller-Tyszkiewicz et al. [34] carried out a longitudinal study and explored the data across three different periods, demonstrating possible changes in the response over time and pointing out the need for more longitudinal evaluations to establish their veracity over time. Similarly, the outcomes of Meireles et al. [14] highlight the need for further research that longitudinally assesses the gestational period and the changes that occur during it.

Regarding the gestational period, nine studies were concerned with evaluating the three gestational trimesters [3,11,24,25,26,27,30,31,32]. Ruble et al. [25] also analyzed postpartum data. One study examined the second and third trimesters of pregnancy [28], and two studies were concerned with only one specific gestational period; Tsuchiya et al. [33] evaluated the second trimester, while Mudra et al. [29] assessed the third quarter. Only one study was not concerned with investigating the gestational period [34]. Specifying the gestational period would further restrict the use of the tool for the population; on the other hand, a pregnant woman undergoes changes in each trimester, each of which has a fundamental characteristic relating to the psychological domain. It is considered ideal to encompass all gestational periods, describing the uniqueness of each one, or to apply the study longitudinally.

As suggested by Morgado et al. [36] and Swami and Barron [37], the instrument’s reliability is one of the criteria to be evaluated. Ways to measure reliability include the evaluation of internal consistency and stability [38]. The vast majority of studies used Cronbach’s α to gauge internal consistency [3,11,24,25,26,27,28,29,30,31,32,34], while the authors of [3] also used McDonald’s ω in their analyses. In addition, the test–retest consists of applying the same test at different times and judging the correlation between two moments [39]. Of the 13 selected studies, only four harnessed this analysis to compare the temporal correlation [3,11,24,34]. Tsuchiya et al. [33] did not use any reliability measure. Various authors recommend including multiple reliability analysis techniques to give greater credibility to the instrument being tested [21,35,36,37]. Hence, it is worth mentioning the work of [3,11,24,34], who employed more than one reliability test, thus bringing more reliability to the tests presented.

As for psychometric analyses, Pasquali [40] pointed out that validity tests seek to assess whether what is intended to be measured is actually being measured. Among the 13 studies analyzed, seven performed EFA and CFA [3,11,24,25,26,29,31], four performed EFA only [27,28,30,32], one performed only CFA [34], and one study did not use any of the analyses [33]. Cash and Smolak [18] indicated that the scientific value of research is related to the quality of the measurement instruments used. As such, the more validity indicators presented in validation and development research, the better the instrument available in the literature will tend to be and, consequently, the lower the risk of measurement errors. As the results show, there were more studies on the creation of scales, and hence, more than one validity test was used (EFA and CFA), which demonstrates researchers’ concern with the quality of the instruments involved.

Another point of analysis was the performance of convergent validation and which instruments were used for this. Among the studies analyzed, eight carried out such validation [3,11,24,26,29,30,31,32]. In all, 21 instruments were used for convergent validation: Body Cathexis Scale (BCS), Quality of Marriage Index (QMI), Hospital Anxiety and Depression Scale (HAD), Multidimensional Assessment of Interoceptive Awareness (MAIA), Mindful Attention Awareness Scale (MAAS), Body Appreciation Scale (BAS), Rosenberg Self-esteem Scale (RSES), Beck’s Depression Inventory (BDI), Cambridge Worry Scale (CWS), State/Trait Anxiety Inventory (SAI-S/T), Social Phobia Inventory (MINI-SPIN), Generalized Anxiety Disorder Scale (GAD), Edinburg Postnatal Depression Scale (EPDS), Body Shape Questionary (BSQ), Eating Disorder Examination Questionnaire (EDE-Q), Sense of Body Boundaries Survey (BBS), Positive and Negative Affect Schedule (PANAS), Satisfaction with Life Scale (SWLS), BAQ, in addition to the BBS subscale, the Experience of Shame Scale (ESS) Body Shame Subscale, and a single-item scale: “How would you define your current physical health status?”. Among the most used constructs and instruments, the EPDS for depression stands out, used four times [24,29,30,31], the RSES for self-esteem, used three times [24,26,30], and the BCS for (dis)satisfaction with one’s body parts and functions [3,11], the GAD-7 for anxiety [29,30], and the BAQ for body dissatisfaction [24,32], each used in two studies. Convergent validity is a type of construct validity that aims to analyze whether there is similarity between measures of constructs that, in theory, are related [35]. Swami and Barron [37] added that it is essential to examine the convergence between the instruments to see if the scores indeed assess what they intend to measure. Therefore, researchers who perform such analyses tend to derive outcomes with better psychometric qualities since they performed comparisons with measures already used on the population.

Specifically, in investigating the Brazilian context, only two studies presented results for pregnant Brazilian women with an instrument created for pregnant women [26] and an instrument adapted from another language [27]. Meireles et al. [26] created the SAS-PW, an instrument composed of 10 items and two subscales (body acceptance and acceptance of pregnancy), whose objective is to determine the self-acceptance of pregnant women during pregnancy. Oliveira et al. [27] carried out the cultural adaptation of the BIPS—originally created for the US [24]—for Brazil. The BIPS is a multifaceted instrument whose main objective is body dissatisfaction. In the Brazilian version, the instrument has 36 items divided into six factors: concern with one’s physical appearance, dissatisfaction with aspects related to body strength, dissatisfaction with one’s skin, attractiveness, prioritization of appearance over function, and dissatisfaction with one’s body parts. Hence, since the diversity of dimensions is critical for the assessment of body image, more specific tools are needed to evaluate different aspects of pregnant women’s body image in the Brazilian population.

The strong point of this work lies in identifying valid instruments used to assess pregnant women’s body image, as well as the dimensions and main characteristics of each study. However, some limitations must be noted. We only searched for texts in English and Portuguese, which may have limited the results and excluded studies that may have been developed in other languages. New searches should include other languages to analyze linguistic and cultural diversity. Furthermore, we did not examine whether the domains of the constructs followed the methodological steps of creating or adapting scales; thus, it is not possible to state whether the domain of the construct evaluates exactly what it proposes to assess. New research must be carried out to verify whether the methods of creating the scales were deductive or inductive, what the methodological steps were for the scale adaptation process, and if these procedures were performed satisfactorily.

Another limitation is that only validity and reliability tests of the studies were performed. Since the quality of the measurement instruments is related to the quality of the instruments used, we suggest that an in-depth analysis of the psychometric procedures be carried out to confirm the quality of the studies involved. In addition, we did not explore the items of the evaluated scales. Morgado et al. [36] pointed out that there may be a limitation in the quality of the written items, which may be ambiguous or difficult to understand and answer. For future studies, we recommend that the scales be evaluated in terms of the wording of their items to reduce comprehension bias.

## 5. Conclusions

We examined 13 articles in this systematic review. Researchers interested in assessing body image in pregnant women should know that the instruments available are BAQ, BUMPS, SAS-PW, PRAQ-R2, BIPS, Prenatal Body Image Questionnaire, Body Image Concern during Pregnancy Scale, Body Experience during Pregnancy Scale, CAQ, and Figure Rating Scale. Australia, Brazil, Canada, Germany, Israel, Iran, Japan, Turkey, the UK, and the USA and have assessment tools to measure some aspect of body image, such as body dissatisfaction, attractiveness, concerns about fat or weight gain, concerns related to physical appearance, pregnant appearance, body and facial characteristics, acceptance of pregnancy, and body satisfaction.

Researchers should take in consideration the objective of the study and the cultural context where the instrument is going to be applied to decide which one would fit properly. We recommend that futures studies cross-culturally adapt and evaluate psychometric characteristics of these instruments for different countries and languages.

An adequate understanding of body image in pregnant women could support health professionals to implement strategies to promote a healthier pregnancy for mothers and babies.

## Figures and Tables

**Figure 1 ijerph-20-02258-f001:**
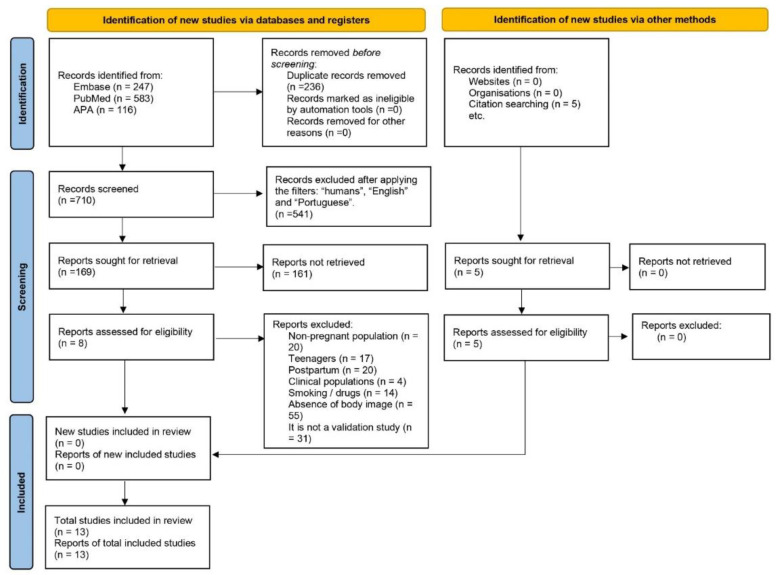
Selection flow diagram of the included studies.

**Table 1 ijerph-20-02258-t001:** Formulation of the search strategy based on PICO.

Population	Intervention	Comparation	Outcomes
Pregnancy	Body image	-	Psychometrics
((Pregnant Women) OR (Woman, Pregnant) OR (Pregnancy))	((Body image) OR (Positive Body image) OR (Body satisfaction) OR (Body Images) OR (Image, Body) OR (Body Identity) OR (Identity, Body) OR (Body Representation) OR (Representation, Body) OR (Body Schema) OR (Body Schemas) OR (Schema, Body) OR (Concept, Self) OR (Self-Perception) OR (Self-Perceptions) OR (Self Perception) OR (Perception, Self) OR (Perceptions, Self) OR (Self Perceptions) OR (Self Confidence) (Confidence, Self) OR (Self Esteem) OR (Esteem, Self) OR (Self Esteems))	-	((Psychological Tests) OR (Psychometrics) OR (Psychometric) OR (Validation Studies) OR (Validation study) OR (Psychologic Tests) OR (Psychologic Test) OR (Test, Psychologic) OR (Tests, Psychologic) OR (Tests, Psychological) OR (Psychological Test) OR (Test, Psychological))

**Table 2 ijerph-20-02258-t002:** Instruments for assessing pregnant women’s body image and their main characteristics.

Reference	Country	Instrument	Type	Number of Items	Aspect of the BI Evaluated	Sample Information	Factor Structure	Internal Consistency	Convergent Validity	Tests	Quality Analysis (N/%)
Fuller-Tyszkiewicz et al. [34]	Australia	Body Attitudes Questionnaire (BAQ)	Validation	44	Body dissatisfaction *Subscales*: feeling fat; strength and fitness; weight/shape salience; attractiveness	176 pregnant, 148 nonpregnant; pregnant: 30.77 years old (SD: ±4.31); nonpregnant: 27.06 years old (SD: ±6.24)	CFA	Cronbach’s α Test–retest (T1; T2; T3 at 8 week intervals)	-	-	35/83.3
Kirk; Preston. [3]	UK	Body Understanding Measure for Pregnancy Scale (BUMPs)	Development	19	Body dissatisfaction *Subscales*: satisfaction with appearing pregnant; concerns about weight gain; physical burdens of pregnancy	378 pregnant EFA: 31.5 years old (SD: ±4.78); gestational period: 26.86 weeks (SD: ±9.33); trimesters: 1st: 12%; 2nd: 35%; 3rd: 53% CFA: 32.28 years old (SD: ±4.37); gestational period: 25.99 weeks (SD: ±9.13); trimesters: 1st: 12%; 2nd: 41%; 3rd: 47%	EFA CFA	Cronbach’s α McDonald’s ω Test–retest	Body Cathexis Scale; relationship satisfaction; anxiety and depression; antenatal attachment; interoception	BCS; QMI; HAD. MAIA; MAAS	39/92.8
Meireles et al. [26]	Brazil	Self-Acceptance Scales for Pregnant Women (SAS-PW)	Development	10	Self-acceptance *Subscales*: body acceptance; pregnancy acceptance	1288 pregnant; EFA: 800 pregnant; CFA: 454 pregnant; 29 years old (SD: ±4.77); trimesters: 1st: 11.8%; 2nd: 39.6%; 3rd: 48.6%	EFA CFA	Cronbach’s α	Body appreciation; self-esteem; depression	BAS; RSES; BDI	41/97.6
Mudra et al. [29]	Germany	Pregnancy-related Anxiety Questionnaire-Revised 2 (PRAQ-R2)	Validation	10	Anxiety *Subscale*: concerns about one’s own appearance	360 pregnant; 32.75 years old (SD: ±3.77); Gestational period: 38.59 months (SD: ±1.81); 3rd trimester	EFA CFA	Cronbach’s α	Pregnancy-specific worries; general state and trait anxiety; symptoms of social phobia; symptoms of generalized anxiety disorder; depressive symptoms	CWS; STAI-S/T; Mini-SPIN; GAD; EPDS	40/95.2
Nagl et al. [30]	Germany	Body Image in Pregnancy Scale (BIPS-G)	Validation	32	Body image *Subscales*: preoccupation with one’s appearance; dissatisfaction with strengths-related aspects; dissatisfaction with one’s body parts; dissatisfaction with one’s complexion; prioritization of appearance over function; concerns about sexual attractiveness	291 pregnant; 31.26 years old (SD: ±4.17); Gestational period: 26.08 weeks (SD: ±9.77); Trimesters: 1st: 14%; 2nd: 30%; 3rd: 56%	EFA	Cronbach’s α	Body dissatisfaction; depression; anxiety; self-esteem; eating disorder psychopathology	BSQ; EPDS; GAD; RSES; EDE-Q.	37/88.1
Oliveira; Carvalho; Veiga. [27]	Brazil	Body Image in Pregnancy Scale (BIPS)	Validation	36	Body image *Subscales*: preoccupation with one’s appearance; dissatisfaction with strengths-related aspects of one’s body; dissatisfaction with one’s complexion; attractiveness; prioritization of appearance over function; dissatisfaction with one’s body parts	180 pregnant; 29.89 years old (SD: ±5.63); Trimesters: 1st: 14.4%; 2nd: 38%; 3rd: 47.6%	EFA	Cronbach’s α	-	-	34/80.9
Talmon; Ginzburg. [31]	Israel	Body Experience during Pregnancy Scale	Development	28	*Subscales*: body agency; body estrangement; body visibility	796 pregnant EFA: 423 pregnant 30.82 years old (SD: ±4.64); Trimesters: 1st: 11.8%; 2nd: 28.2%; 3rd: 60%; CFA: 373 pregnant 30.58 years old (SD: ±4.74); Trimesters: 1st: 9.7%; 2nd: 34.6%; 3rd: 55.8%	EFA CFA	Cronbach’s α	Disrupted body boundaries; body shame; positive and negative affect; life satisfaction; depression; self-rated health	BBS; ESS; PANAS; SWLS; EPDS; direct question: “How would you define your current physical health status?”	36/85.7
Tsuchiya; Yasui; Ohashi. [33]	Japan	Figure Rating Scale (FRS)	Development	2	Body dissatisfaction	161 pregnant; 33.2 years old (SD: ±3.68); Gestational period: 22 weeks (SD: ±1.56) 2nd trimester	-	-	-	-	30/71.4
Uçar et al. [28]	Turkey	Body Image Concerns during Pregnancy Scale	Development	23	Body image concerns during pregnancy *Subscales*: avoidance and social concerns; concerns about weight gain; concerns about the future; concerns about one’s physical appearance	320 pregnant; 27.5 years old (SD: ±4.7); Gestational period: 28.8 weeks (SD: ±7.2); 2nd and 3rd trimesters	EFA	Cronbach’s α	-	-	35/83.3
Watson et al. [24]	USA	Body Image in Pregnancy Scale (BIPS)	Development	36	Body image *Subscales*: preoccupation with one’s physical appearance; dissatisfaction with physical strength; dissatisfaction with facial features; sexual attractiveness; prioritizing physical appearance over body functions; appearance-related behavioral avoidance; dissatisfaction with one’s body parts	251 pregnant; 30 years old (SD: ±4.34); Gestational period: 24.22 weeks (SD: ±8.58); 1st, 2nd, and 3rd trimesters	EFA CFA	Cronbach’s α Test-retest	Body attitudes; self-esteem; depressive symptoms	BAQ; RSES; EPDS	38/90.4
Satir; Hazar. [11]	Turkey	Body Understanding Measure for Pregnancy Scale (BUMPs)	Validation	17	Body dissatisfaction *Subscales*: concerns about weight gain and physical difficulties; dissatisfaction with appearing pregnant	265 pregnant; 28.73 years old (SD: ±4.40); Gestational period: 23.49 weeks (SD: ±9.97); Trimesters: 1st: 19.6%; 2nd: 42.3%; 3rd: 38.1%	EFA CFA	Cronbach’s α Test–retest	Body Cathexis Scale	BCS	40/95.2
Sohrabi; Kazemi; Farajzadegan. [32]	Iran	Prenatal Body Image Questionnaire (PBIQ)	Development	30	Female body ideals *Subscales*: fitness and beauty; lower body fat; attention to changes in pregnancy; shame; sexual attractiveness; negative feelings about skin changes; the symbol of motherhood	300 pregnant; 28.1 years old (SD: ±4.9); Gestational period: 26.4 weeks (SD: ±6.9); Trimesters: 1st: 5%; 2nd: 41%; 3rd: 54%	EFA	Cronbach’s α	Body Attitudes Questionnaire	BAQ	39/92.8
Ruble et al. [25]	Canada and the USA	Childbearing Attitudes Questionnaire (CAQ)	Development	73	*Subscales*: maternal worries; maternal self-confidence; relationship with one’s husband; relationship with one’s mother; body image; identification with pregnancy; feelings about children; negative self-image; attitude toward breastfeeding; pain tolerance; interest in sex; denial; negative aspects of caretaking; feelings of dependency; social boredom; information seeking	667 pregnant; 29 years old; Planned to get pregnant: 17%; Trimesters: 1st: 8.9%; 2nd: 17.9%; 3rd: 17.5%; 1st month postpartum: 19.3%; 3rd month postpartum: 19.4%	EFA CFA	Cronbach’s α	-	-	34/80.9

Legend: EFA = exploratory factor analysis; CFA = confirmatory factor analysis; BCS = Body Cathexis Scale; QMI = Quality of Marriage Index; HAD = Hospital Anxiety and Depression Scale; MAIA = Multidimensional Assessment of Interoceptive Awareness; MAAS = Mindful Attention Awareness Scale; BAS = Body Appreciation Scale; RSES = Rosenberg Self-Esteem Scale; BDI = Beck’s Depression Inventory; CWS = Cambridge Worry Scale; STAI S/T = State/Trait Anxiety Inventory; Mini-SPIN short version = Social Phobia Inventory; GAD = Generalized Anxiety Disorder Scale; EPDS = Edinburg Postnatal Depression Scale; BSQ = Body Shape Questionnaire; EDE-Q = Eating Disorder Examination Questionnaire; BBS = Sense of Body Boundaries Survey; ESS = Experience of Shame Scale; PANAS = Positive and Negative Affect Schedule; SWLS = Satisfaction with Life Scale; BAQ = Body Attitudes Questionnaire; USA = United States of America.

## Data Availability

Not applicable.

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
