# Peer review of "Body Image Assessment Tools in Pregnant Women: A Systematic Review"

_ijerph, 2023, doi:10.3390/ijerph20032258_

Round 1

Reviewer 1 Report

The authors have done an extensive literature search regarding body image assessment tools in pregnant women. The manuscript is well written where methods are clearly described and the discussion is thorough. I just have one comment about the conclusion which should be addressed. In the conclusion, the authors stated that "Researchers interested in this target audience should use the instruments identified here to assess pregnant women’s body image in different countries and languages, allowing for a global evaluation". Their conclusion should be more specific, as in what instrument they recommend and why. If it has some limitations how it can be improved. They should emphasize what future studies should follow and how should they be designed better. They can include a table stating pros and cons of each instrument and its applicability. This will make the manuscript stronger and will help the researcher to make educated decisions for further studies, signifying their work. 

Reviewer 2 Report

Dear Authors, 

Thank you for preparing this manuscript. Overall the review is well-conducted and results are written clearly. 

Please find below some suggestions. 

Introduction

Is there any number of pregnant women affected by body dissatisfaction? Is this body dissatisfaction linked with poor nutrition practices and effects on the developing foetus? 

It would be nice to give this information as a rationale for the importance of tools for body assessment. 

Selection of studies

Did you put any limit in terms of year of publication? 

You say you selected the filter for 'full text available'. Were the full-texts available linked with your institute's subscriptions, or are the papers you retrieved free to everyone? 

Why did you choose English and Portuguese articles? Any explanation for this choice? 

Results

More results need to be added to this section. You do not mention anything about the tools identified in the review, which is the main aim of the review. I can see there is a section on this in the discussion. You need to transfer that to the results section. 

Did you evaluate the quality of the included studies? 

Discussion

You need to move some of the discussion into the results section. 

Reviewer 3 Report

Overall this is a useful review of instruments assessing body image evaluation in pregnancy.

There are minor issues that needs attention:

1)  throughout all paper Authors fail to spell in full acronyms used the first time they appear. This is very annoying and should be fixed 

2) throughout all paper Authors replaced first Author of a paper with the number, making the paper illegible. For example in page one line 43 the sentence begins with [12] discovered Please always mention the first Author surname adding et al. if more than two Authors in this and other alike cases

3) The definition of pregnancy in the very first lines of the paper is plainly wrong since many pregnancies alas DO NOT culminate in the baby's birth. This definition not only is wrong but utterly unsensitive to women whose pregnancies ended in a termination, spontaneous or not. 

4) Search should be made also in SCOPUS and Web of Science, why these important databases have been ignored ?
